# Intracardiac Porcupine Quill Migration in a Dog: Multimodality Imaging Findings and Surgical Management

**DOI:** 10.3390/vetsci9120700

**Published:** 2022-12-16

**Authors:** Antonello Bufalari, Giulia Moretti, Eleonora Monti, Lisa Garofanini, Giovanni Angeli, Francesco Porciello, Maria Chiara Marchesi, Domenico Caivano

**Affiliations:** Department of Veterinary Medicine, University of Perugia, Via San Costanzo 4, 06126 Perugia, Italy

**Keywords:** computer tomography, heart, *Hystrix cristata*, migrating foreign body, porcupine quill, thoracic surgery, ultrasound

## Abstract

**Simple Summary:**

Porcupine quill injuries are common in dogs; however, intracardiac quill migration is rare. Considering the high rate of complications secondary to quill penetration and migration, veterinarians should be aware of risks associated with long-standing foreign bodies even when no clinical signs are evident. A dog was referred for three episodes of recurrent fever that resolved with antibiotic therapy. Ultrasonography and computed tomography revealed a single linear hyperechoic structure traversing the interventricular septum from the heart base to the cardiac apex. A median sternotomy was performed, and a porcupine quill was removed under a direct ultrasound guide. The dog recovered uneventfully. Dogs without evidence of severe cardiac injury secondary to a single intracardiac quill may have a good prognosis.

**Abstract:**

The porcupine is a nocturnal quill-bearing rodent common throughout Italy. When threatened, it defends itself by erecting its quills, rattling its tail, and running sideways or backyard into predators. The quills are released upon contact with assailants and tend to migrate into several body tissues or cavities due to the unique inverted cuticles (crenate) directed downwards to the quill base (a sort of “harpoon effect”). Intracardiac migration of porcupine quills has been reported in a few dogs with severe clinical signs. This case report describes a single porcupine quill [*Hystrix cristata*, (*HC*)] migration through the heart in a dog and the use of multimodality imaging for the diagnosis and surgical approach. A 4-year-old 25 kg female mixed-breed dog was presented for evaluation of recurrent fever. Transthoracic echocardiographic examination showed a linear, hyperechoic foreign body traversing the interventricular septum from the heart base to the cardiac apex. Computed tomography and transesophageal echocardiography confirmed the presence of a single structure resembling a quill; a later esophageal endoscopy showed no anatomical alterations or mucosal injuries. Following median sternotomy and after accurate localization by intraoperative ultrasound, the quill was successfully removed. The dog had a good recovery without evidence of severe cardiac injury secondary to an intracardiac quill. To the best of the authors’ knowledge, the identification of an intracardiac *HC* quill by ultrasonography and CT and its successful removal by ultrasound guide has not been previously reported.

## 1. Introduction

The European porcupine (*Hystrix cristata*) is a quite common rodent that lives in Italy, mainly in wooded and bushy areas. The antipredator behavior of these rodents is quite fascinating: tail rattling, in most cases, is sufficient to move away solitary predators, while backyard/sideways attacks are exhibited only in extreme situations of direct strikes [1]. Porcupines do not throw their quills; instead, they contract muscles and lift their quills toward the attackers, and since the quills are loosely rooted, they are released upon contact and remain embedded in the assailants [1]. *HC* quills have a cylindrical shape in the lower two-thirds of the shaft and then flatten toward the apex; they can reach a significant length of 380–400 mm, while the thickness is about 2 mm. Moreover, there are up to 12 furrows on the shaft, and they correspond to the *septa* dividing the thick alveolar medulla [2]. The features of these quills differ significantly from those of the American porcupine (*Erethizon dorsatum*), are significantly shorter (85 mm), smooth, weakly bent, and slightly flattened; the tip is barbed (microscopic backward-facing deployable barbs), and the alveolar medulla is strongly developed but is not divided by thick *septae* [2,3]. The tip of the quill of *Hystrix* spp. Is sharp and not barbed, whereas the external end has a unique inverted cuticle directed downwards to the quill base (like a “harpoon”) [2]. These specific properties (along with the barbed tip in the *Erethizon dorsatum*) reduce the possibility of pulling out the quills and aids penetration and migration through the tissues [1,2,3]. Porcupine quills can be contaminated by bacteria, and they directly irritate soft tissues, causing a septic foreign body reaction [4].

Porcupine quills are frequent foreign bodies retrieved in canine bodies, especially in the head, neck, and thorax [5,6,7,8,9,10,11,12,13]. Quills from *Hystrix* and *Erethizon*, frequently break off at the skin level, thus hampering their identification at the time of patient presentation for veterinary care [9]. Early diagnosis and subsequent complete removal improve the prognosis and outcome [8]; if this is not possible, quill migration and, consequently, injuries in several body tissues or cavities can occur. Migration to the thorax usually causes pleural effusion and pneumothorax [8]. When the eyes are struck, the collateral damages are severe and can lead to loss of vision [6,12]. In case of migration to the joints, it results in septic arthritis and secondary endocarditis [7]. Some reports indicate a possible migration to the vertebrae, spinal canal, and brain causing progressive neck pain, severe encephalitis, and, eventually, death [11,14,15]. Although quill migration is common in dogs, migration into the heart is rare [9,16]. Quills may penetrate the heart directly (transthoracic penetration) or indirectly by migration from the distal cervical portion into the thoracic cavity [8]. The degree of heart injury impacts the timing of presentation and diagnosis, influencing the management strategy. If the foreign body rapidly migrates through the thorax, cardiac tamponade, hemothorax, and pneumothorax can occur [13]. Therefore, determining the best diagnostic imaging modalities to localize the quill is essential. The exact position of the foreign body provides useful information on the possible complications and can help to ensure complete removal.

In this report, a case of porcupine quill migration through the heart in a dog with recurrent fever and the use of multimodality imaging for the diagnosis, as well as its removal under a direct ultrasound guide, has been described.

## 2. Case Presentation

### 2.1. Case Description and Clinical Investigations

A 4-year-old female, 25 kg, mixed breed, was presented to the Veterinary Teaching Hospital of the University of Perugia for evaluation of recurrent fever. Four months prior to the presentation, the dog had been evaluated by the referring veterinarian for swelling and pain in the ventral region of the neck associated with a presumptive snake bite and had been treated with antibiotic/corticosteroid therapy. Since this event, the dog has presented three episodes of recurrent fever resolved with antibiotic therapy. No other clinical signs were reported by the owner and referring veterinarian. At the time of presentation to our Hospital, the dog showed lethargy, fever (40.4 °C), and mild tachypnoea (28 breaths/min). A complete blood count and serum chemistry were performed. Abnormalities included a neutrophilic leukocytosis (19.43 × 10^3^/µL, reference range 6–12 × 10^3^/µL), a mild hyporegenerative anemia (RBC 4.05 × 10^6^/µL, reference range 5.5–8.21 × 10^6^/µL; Hb 10.1 g/dL, reference range, 12 to 18 g/dL; Hct 29.2%, reference range, 37 to 55%; MCV 72.1 fL, reference range 60–72 fL; MCH 24.9 pg, reference range 20–25 pg; MCHC 34.6%, reference range 32–39%; RDW 15.2%, reference range 12–16%) and elevated alkaline phosphatase (198 U/L, reference range 10 to 100 U/L).

Thoracic radiographs (Figure 1) and abdominal ultrasound were performed, and no abnormalities were observed. The electrocardiographic exam showed no cardiac arrhythmias.

To rule out infective endocarditis or myocarditis, an echocardiographic examination was performed using an ultrasound unit equipped with a multifrequency 1–4 MHz phased-array transducer (MyLab^TM^ Eight, Esaote, Genova, Italy) and a linear hyperechoic structure traversing the interventricular septum from the heart base to the cardiac apex was visualized. In the longitudinal plane, the foreign body appeared as two well-defined mural parallel hyperechoic lines, with multiple additional hyperechoic lines parallel to the walls (Figure 2). In the transverse plane, the foreign body appeared as a circular hyperechoic structure with distal acoustic shadowing (Figure 2). A hypoechoic zone surrounded the foreign body at the heart base (Figure 2). No other cardiac anomalies were observed. A Computed tomography (CT) examination of the thorax was performed with a 16-slice helical CT scanner (Fujifilm FCT Speedia, Fujifilm Italia S.p.a., Milan, Italy) before and after intravenous contrast administration. The acquisition parameters were 120 kV, 100 mAs, pitch 1, rotation time tube 0.75 s, slice thickness 2.5 mm reformatted with 1.25 mm of overlapping. The images were studied with a soft tissue algorithm in axial and multiplanar views. Many volumetric reconstructions were useful for the correct localization and extension evaluation of the foreign body. CT revealed, in axial view, the presence of a quite circular area, mildly hyperdense with respect to the cardiac muscle, that was visible in a consecutive sequence of images starting approximatively from the heart base to the surface of the left side of the cardiac apex across the interventricular septum. Multiplanar reconstructions were obtained following the long axis of the foreign body in order to define the shape and the dimension, which was estimated at around 80 mm for the length and 5 mm for the width (Figure 3). Moreover, an elongated triangular structure with hyperdense borders and a hypodense core was possible to observe. The volumetric reconstruction is defined as an elongated structure that appears as a pointed conical object.

Esophageal endoscopy was performed to rule out esophageal migration of the foreign body, and no anatomical alterations or mucosal injuries were observed. Transesophageal echocardiography, performed while the dog was under general anesthesia for endoscopy, confirmed the transthoracic echocardiographic findings (Figure 4). A diagnosis of intracardiac foreign body, suggestive of migrating quill, traversing the interventricular septum without entering the cardiac chambers was performed, and surgical removal of the foreign body was recommended.

### 2.2. Surgical Approach and Outcome

After formal consent of the owner, the patient underwent general anesthesia, using a standard clinical protocol: fasting for 12 h and free access to water; premedication was performed with methadone (0.2 mg/kg IV; Semfortan, Dechra Pharmaceutics, Northwich, UK); 10 min after premedication, anesthesia was induced with propofol (4 mg/kg IV; PropoVet, Zoetis Italia S.r.l., Rome, Italy) and midazolam (0.2 mg/kg IV; Midazolam Accord Healthcare, Accord Healthcare Italia S.r.l., Milan, Italy) through a cephalic venous cannula previously and aseptically placed. Following tracheal intubation, the dog was connected to a rebreathing circuit (50 mL/kg oxygen flow) and maintained with isoflurane (1.2–2.0%; Isoflo, Zoetis Italia S.r.l, Rome, Italy) in 100% oxygen. The patient was mechanically ventilated during anesthesia. Lactated Ringer solution (10 mL/kg/h CRI) was infused during the entire anesthetic procedure. Meloxicam (0.2 mg/kg, SC; Metacam, Boehringer Ingelheim Italia S.p.a., Milan, Italy) and cefazolin (30 mg/kg IV; Cefazolina Teva, Teva Italia S.r.l., Assago, Italy) were administered during patient preparation and at least 120 min prior to surgery. After the routine aseptically preparation of the operative field, a ventral midline sternotomy was performed. Intraoperative echocardiography was used to evaluate the localization of the foreign body and confirm its position through the interventricular septum. The pericardium did not show any modification. A small pericardiectomy (4 × 4 cm) was carried out to provide a clearer exposure of the epicardium wall. A small bulging round area was noted at the apex of the left ventricular wall; however, the quill was not yet visible on the cardiac surface. An ultrasound probe, encased in a sterile protective cover, was placed over the cardiac apex and intraoperative ultrasonography allowed to visualize the quill and its tip. A double horizontal mattress suture tension (3-0 polypropylene) was placed in the epicardium wall around the area of the tip of the quill (at about 10 mm from the apex of the heart). Under ultrasonographic direct visualization, a small (8 mm) incision of the ventricular wall was carried out by n. 11 scalpel blade; the depth of the incision was about 6 mm; through the incision, a small, curved tip hemostatic forceps were introduced, and under ultrasound guide, the quill was grasped and gently pulled out from the heart (Figure 5, Appendix A). No cardiac arrhythmias were recorded during the removal of the quill. A swab of the area housing the quill was performed, and the double sutures were tightened to prevent hemorrhage. The swab, as well as the quill, were submitted for aerobic and anaerobic culture. Intraoperative ultrasonographic examination performed after the removal did not visualize any fragmented parts of the extracted quill. The thoracic cavity was explored to rule out lung lobes damage. Before closure, the chest cavity was flushed with a lukewarm sterile saline solution and inspected for air leakage from the lung lobes. A 12 French gauge thoracostomy tube was inserted in the left hemithorax. The thoracic cavity was closed with 18-gauge stainless steel wires placed around the *sternebrae* in an X-cross interrupted pattern. The pectoral muscles were closed with 0 polydioxanone sutures in a cruciate mattress pattern technique. The subcutaneous tissues were closed with a simple continuous pattern of 3-0 polydioxanone, and the skin was closed with an intradermic simple continuous pattern of 4-0 poliglecaprone. Following surgery, the dog was weaned from mechanical ventilation without complication. Recovery from anesthesia was uneventful, with extubation 15 min after inhalant anesthesia was discontinued. The thoracostomy tube was removed 24 h later. Initial treatment post-operatively included cefazolin (30 mg/kg IV q12 h; Cefazolina Teva, Teva Italia S.r.l., Assago, Italy), lactated Ringer’s solution (2 mL/kg/h IV), and meloxicam (0.1 mg/kg SC q24 h, Metacam, Boehringer Ingelheim Italia S.p.a., Noventana, Italy). For pain management, the initial treatment included methadone (Semfortan, Dechra Pharmaceutics, Northwich, UK) 0.2 mg/kg IM q4 h. Two days post-operatively, results from the cultures revealed the growth of an amoxicillin-resistant *Pseudomonas aeruginosa* with sensitivity to amikacin, gentamicin, enrofloxacin, and marbofloxacin. Therefore, antibiotic therapy was changed to enrofloxacin (5 mg/kg PO q24 h for six weeks; Baytril Flavour, Elanco Italia S.p.a, Sesto Fiorentino, Italy). A few days after the surgery, a surgical site infection (SSI) of the wound involving pectoral muscles was noted and treated by surgical curettage of the sternal area, local swab for antimicrobial testing, and abundant lavage. Two days later, preliminary culture results from the infected wound indicated growth of *Pseudomonas aeruginosa* that was only sensitive to clindamycin, amikacin, gentamicin, enrofloxacin, and marbofloxacin. Amikacin (10 mg/kg; Amikacina Teva, Teva Italia S.r.l., Assago, Italy) was infiltrated locally three times a week. The wound improved, and the draining tract resolved, so the dog was discharged after four weeks after the surgery.

Follow-up six months after the surgery revealed that the dog was working to full capacity without evidence of any clinical signs.

## 3. Discussion

To the authors’ knowledge, this is the first clinical case reporting an ultrasound and CT visualization of an intramyocardial *HC* quill and its surgical removal using the intraoperative ultrasonographic guide.

Porcupine quills are unique among migrating foreign bodies, and their injuries involve more areas of the body. Most commonly, the areas involved are represented by the head and neck, followed by limbs and trunk in dogs [8]. In case of missing quills at the first examination, they have a malignant tendency to migrate deeper into tissues rather than backing out due to their specific inverted cuticles. Nevertheless, quills are irritant and contaminated and can cause several types of microlesions [4,13,17]. The identification of small porcupine quills remains difficult with the use of current imaging techniques and modalities [13,18]: CT, magnetic resonance imaging (MRI), and ultrasound have been previously reported for identification of porcupine quills with varying accuracy [7,9,11,15,16,17,18]. Porcupine quills are typically not visible on radiographs [6]. Conversely, porcupine quills show the following ultrasonographic features: two distinct parallel hyperechoic lines that converge at the tip of the quill associated with multiple additional echogenic lines parallel with the walls when the quill is parallel to the transducer. When the porcupine quill is transverse to the transducer can be easily misinterpreted as small blood vessels and can appear as small hyperechoic tubular structures with or without acoustic shadowing [6,7,11,15]. Previous studies have documented the effectiveness of ultrasound for the detection of quills in the eye, joints, and flexor tendon [7,15]. Transthoracic echocardiography is a valuable imaging modality for detecting quills in the pericardial region or into the heart [8,9,16]. CT is considered the modality of choice for the thoracic cavity because rapid scan times result in decreased motion artifacts, and cross-sectional imaging eliminates superimposition, providing accurate imaging down to 1 mm [19]; however, due to the size and location of quills, motion artifact, residual pneumothorax, atelectasis, and axial thickness, a definitive diagnosis and accurate localization can be a challenge. Moreover, the quill shows a density similar to soft-tissue inflammation associated with its migration [9]. Porcupine quills on the CT can appear as a hyperdense linear structure [11,15]. The use of MRI has been reported in the diagnosis of a quill foreign body in dogs [11,14,17]. However, MRI has poorer spatial resolution than ultrasound and CT, despite its excellent contrast resolution. Moreover, the intensity of the quill and the suspected hemorrhages, being similar, could limit the visualization of the quill. In our case report, transthoracic and transesophageal ultrasound allowed us to identify and localize the porcupine quill. Moreover, intraoperative ultrasound allowed to guide foreign body removal from the heart and to confirm the complete removal of the quill without complications. CT confirmed the ultrasound findings, identifying a single intramyocardial quill. MRI was unavailable in our hospital.

In the veterinary literature, migration of foreign bodies such as grass awn, sewing needles, wooden skewers, metallic projectile, and catheter fragments into the heart has been previously reported [20,21,22,23,24]. In our clinical case, we suppose that the long quill entered the body at the right recess located at the base of the neck through the pectoralis superficialis and brachiocephalic muscles. Then, it migrated through the soft tissues entering the thorax cavity and following the cranial mediastinum, pierced the heart, avoiding the great vessel at the base of the heart. Finally, it stabilized in the interventricular septum without penetrating cardiac chambers and damaging valve structures.

Currently, there are no indications regarding the management of a long myocardial quill in veterinary medicine. Surgical intervention has been debated for small foreign bodies, such as quills, sewing needles, projectiles, and skewers, in humans and animals [13,21,22,25,26]. Most authors recommend early surgical intervention in people with acute injuries because of the life-threatening complications. Moreover, the removal of all intra-chamber foreign bodies has been consistently suggested because of their potential for embolization, whereas myocardial foreign bodies can be monitored and only need removal if the patient becomes symptomatic [22,27]. In veterinary medicine, the ideal treatment of intracardiac migrating porcupine quills remains unclear; the perioperative findings and outcomes of dogs undergoing exploratory pericardiotomy for cardiac quill migration have been previously reported in only a small group of cases [9,13,16]. However, none of these reports includes a description of the use of an ultrasound guide to safely remove foreign bodies inserted into the myocardium. Some reports suggest an exploratory thoracotomy to investigate and examine gross evidence of pericardial injury and, eventually, identify and remove quills [9,13,16]. However, exploratory thoracotomy (lateral or median sternotomy) should be carefully evaluated in absence of specific localization of the foreign body. In our case, the ultrasound effectively played a substantial and meaningful role in identifying an intracardiac foreign body and helped the surgeon during its extraction from the myocardium.

Thoracoscopic exploration and removal of porcupine quills have been previously described in humans; however, postoperatively, complications due to persistent porcupine quill migration were observed [28]. In our report, a median sternotomy from the third *sternebrae* to the *xiphoideus* process was preferred permitted the use of ultrasound for identifying the quill and its tip. Indeed, it would be advisable to grasp the tip of the foreign body and not the tail (because of the inverted cuticle). Moreover, this approach allowed us to pull out the quill with the correct line of extraction since the thorn was positioned along both *septa* following the longitudinal axial of the heart. Left lateral access would not have had the same easy access to retrieve the quill in a straight line because of the presence of the caudal rib. Moreover, the safe removal of the thorn from the heart goes through a correct identification of the quill by ultrasound probe in order to evaluate the direction of the thorn and to determine the correct maneuver for its removal. Since foreign bodies like quills become less rigid and more friable the longer they remain in tissues making them more susceptible to breakage during removal [7], the maneuvers of extraction should be attempted following a straight line with a steady and firm linear motion, avoiding the breakage of the quill due to a different angle of cleaving during extraction.

In our clinical case, the quill was localized into the interventricular septum; in a different scenario, where a quill perforates the atrial or ventricular wall and/or damages the valves, an attempt to remove the foreign body, without performing a cardiopulmonary bypass could lead to catastrophic consequences due to hemorrhage across the septum or systemic air embolism [29]. In our clinical case, cardiopulmonary bypass was not required since the quill did not break the walls or valves but remained in the area of the interventricular septum. A more aggressive approach, such as opening the heart, however, would allow the necessary time for surgical removal of the foreign body and eventual repair of the myocardium or valves [29]. Determination of the exact localization of the quill in the heart by CT and Ultrasound is mandatory to adequately plan the type of surgery and guide the surgeon to ensure its complete and safe removal.

In our specific case, doubt arose about the fact that retrieval of the quill could be challenging since its embedding into the myocardium. Furthermore, the closure of the myocardium could generate some technical difficulties. Thankfully, the retrieval was smooth and easy. The pulling-out effort was minimal, and no embedding issues were experienced. Ultrasound re-checking after removal did not point out any small or fragmented parts of the extracted quill that may become embedded in deeper tissues of the heart. Closure of the epicardium was easy and fast using the same tension sutures previously applied.

Interestingly, despite the fact that the quill migrated through the thoracic cavity, no air leakage or scars were identified from any of the lung lobes during the direct inspection, suggesting that the quill did not perforate any lung lobes.

As mentioned before, a few days after recovery, the dog showed SSI of the caudal part of the wound in the area of the pectoralis muscles. Results of bacteriological culture showed *Pseudomonas aeruginosa* similar to what was revealed from the quill and his housing into the heart. We hypothesize that a passage of infection through the sternum and pectoral muscles occurred from the quill housing into the heart, creating SSI of the wound. *P. aeruginosa* is an opportunistic pathogen that causes a wide spectrum of infections and leads to substantial morbidity in immunocompromised patients. Frequently, *P. aeruginosa* demonstrates resistance to multiple antibiotics, as in our case, jeopardizing the selection of appropriate treatment. In the dog described in our case, the long therapy with enrofloxacin and amikacin locally infiltrated [30] significantly improved the SSI and facilitated wound healing.

## 4. Conclusions

Porcupine quill injuries are common in dogs, although intracardiac migration is rare and is commonly associated with severe clinical signs. Diagnosis of intracardiac porcupine quill migration in dogs without evidence of severe cardiac injury can be a challenge for clinicians. An accurate diagnosis and identification of the quill by a multimodality imaging approach is mandatory for planning a safe removal from the myocardium. Moreover, intraoperative ultrasound is useful in guiding the extraction of the foreign body.

## Figures and Tables

**Figure 1 vetsci-09-00700-f001:**
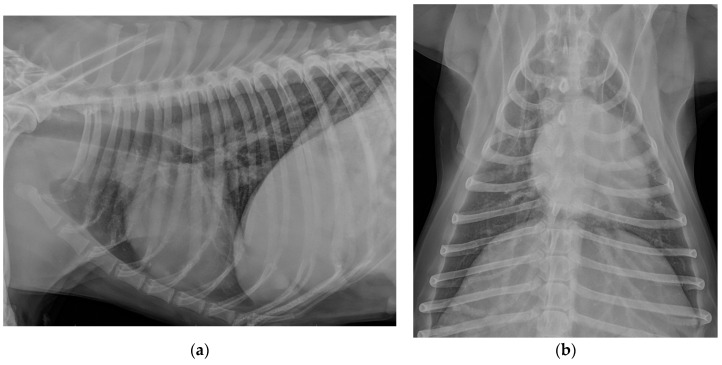
Radiographic examination of the thorax of the dog affected by intracardiac porcupine quill migration. (**a**) Left lateral and (**b**) ventrodorsal projections show no abnormalities.

**Figure 2 vetsci-09-00700-f002:**
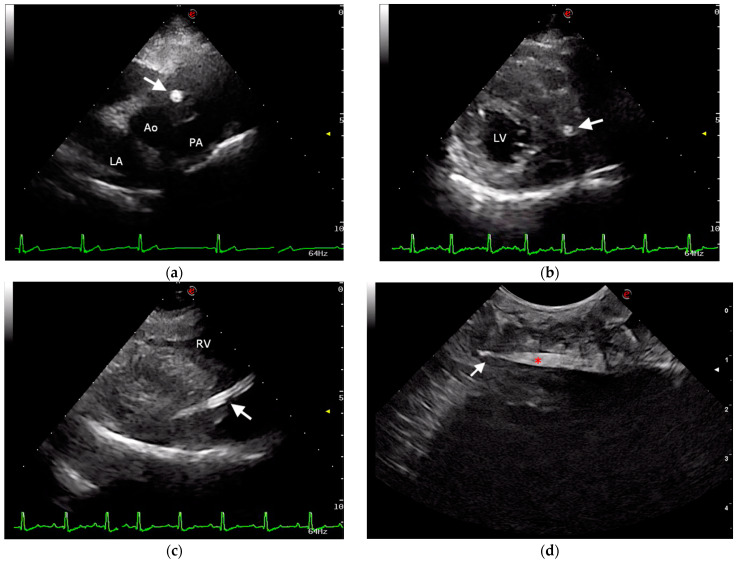
Ultrasonographic images of the intramyocardial porcupine quill. (**a**) Transthoracic echocardiographic image obtained from a right parasternal short-axis view at the level of the heart base showing the foreign body as a circular hyperechoic structure (arrow); (**b**) transthoracic echocardiographic image obtained from a right parasternal short-axis view at the level of the left ventricle showing the foreign body as a circular hyperechoic structure (arrow) with distal acoustic shadowing; (**c**) transthoracic echocardiographic image obtained from a modified right parasternal long-axis view optimized to visualize the foreign body in its longitudinal plane (arrow); (**d**) Intraoperative ultrasonographic image of the tip (arrow) of the quill (red asterisk). Ao, aorta; PA, pulmonary artery; LA, left atrium; LV, left ventricle; RV, right ventricle; e, orientation marker.

**Figure 3 vetsci-09-00700-f003:**
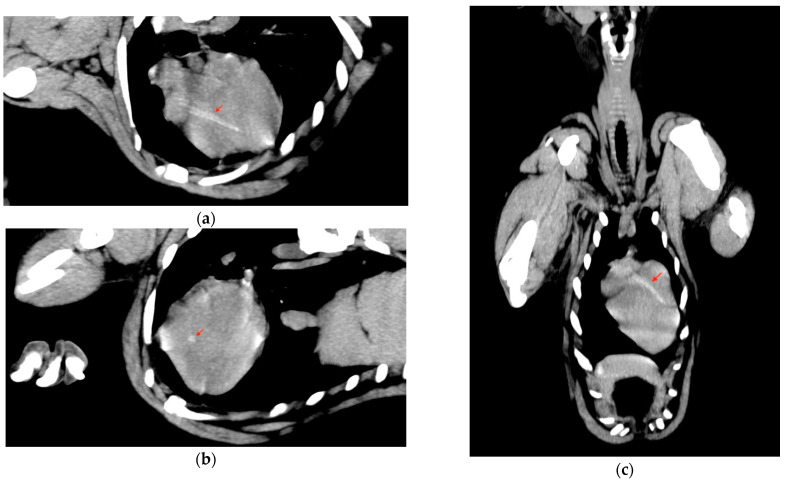
CT multiplanar reconstructions of the cardiac area consent to define the elongated and acuminate shape of a mildly hyperdense intracardiac foreign body (red arrow). (**a**) Oblique cardiac reconstruction shows the length of the foreign body and its tip nearby the external surface of the cardiac wall. The tri-layered aspect allows us to hypothesize the presence of a medullary-like cavity that is confirmed in a para-axial view (**b**) in which is a ring-like structure is visible. (**c**) Dorsal reconstruction allows for the localization of the foreign body in the thickness of the intraventricular septum for most of its length.

**Figure 4 vetsci-09-00700-f004:**
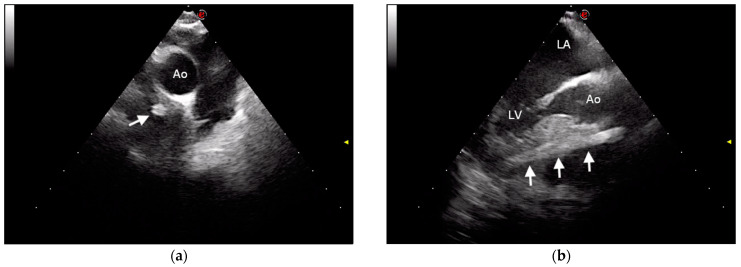
Transesophageal ultrasonographic images of the intramyocardial porcupine quill. (**a**) Transesophageal echocardiographic image of the heart base showing the foreign body as a circular hyperechoic structure (arrow); (**b**) transesophageal echocardiographic image of the left heart chambers and aorta showing the foreign body in its longitudinal plane (arrows). Ao, aorta; LA, left atrium; LV, left ventricle; e, orientation marker.

**Figure 5 vetsci-09-00700-f005:**
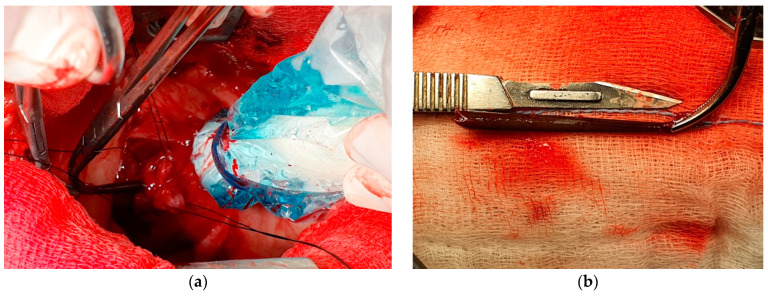
Intraoperative images of the surgical removal of the porcupine quill through median sternotomy. (**a**) Ultrasound-guided removal of the porcupine quill by a curved hemostat from a small incision on the cardiac apex; (**b**) the porcupine quill after the removal.

## Data Availability

The data presented in this study are available in the article.

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
