# Peer review of "Intracardiac Porcupine Quill Migration in a Dog: Multimodality Imaging Findings and Surgical Management"

_vetsci, 2022, doi:10.3390/vetsci9120700_

Round 1

Reviewer 1 Report

I thank the Editor for the opportunity to review this manuscript, which describes the diagnosis and surgical removal of an unusual intracardiac foreign body. The case is well written, as all the procedures used are reported in detail, and provides useful information for clinical practice. I am not a native speaker, but in my opinion English is fine. 

I have only a few minor revisions to suggest to the authors:

1) The authors report that the dog had mild hyporigenerative anemia. In this regard, it would be advisable to add the values of the RBC indices in the text.

2) Since the foreign body passed through the interventricular septum almost for all its length, it might be interesting for the reader to know whether the patient had arrhythmias on electrocardiographic examination at the time of clinical presentation or during surgical removal.

Author Response

We thank the Reviewer for her/his encouraging comments and valuable suggestions. We have addressed all the comments raised and we hope that our changes below will satisfy the Reviewer.

I thank the Editor for the opportunity to review this manuscript, which describes the diagnosis and surgical removal of an unusual intracardiac foreign body. The case is well written, as all the procedures used are reported in detail, and provides useful information for clinical practice. I am not a native speaker, but in my opinion English is fine.

I have only a few minor revisions to suggest to the authors:

1) The authors report that the dog had mild hyporigenerative anemia. In this regard, it would be advisable to add the values of the RBC indices in the text.

We have added the values of the RBC indices following the Reviewer’s suggestion: “…..mild hyporegenerative anemia (RBC 4.05x106/µL, reference range 5.5-8.21x106/µL; Hb 10.1 g/dL, reference range, 12 to 18 g/dL; Hct 29.2 %, reference range, 37 to 55 %; MCV 72.1 fL, reference range 60-72 fL; MCH 24.9 pg, reference range 20-25 pg; MCHC 34.6 %, reference range 32-39 %; RDW 15.2 %, reference range 12-16 %)”.

2) Since the foreign body passed through the interventricular septum almost for all its length, it might be interesting for the reader to know whether the patient had arrhythmias on electrocardiographic examination at the time of clinical presentation or during surgical removal.

We thank the Reviewer for her/his suggestion. We have added the following sentences in the manuscript: “Electrocardiographic exam showed no cardiac arrhythmias.” and “No cardiac arrhythmias were recorded during the removal of the quill.

Reviewer 2 Report

Dear Author, the manuscript is very interesting and well written.

Your case is very unique. Being a case report that underlines the importance of diagnostic imaging, I suggest you add more images to complete the manuscript. It may be of interest to add orthogonal radiographic images of the chest, although no abnormalities were observed.

Please add the model of the ultrasound machine and the probes (frequency) you use in the text.

Also add Transesophageal echocardiography images

Thank you

Author Response

We thank the Reviewer for her/his encouraging comments and valuable suggestions. We have addressed all the comments raised and we hope that our changes below will satisfy the Reviewer.

Dear Author, the manuscript is very interesting and well written.

Your case is very unique. Being a case report that underlines the importance of diagnostic imaging, I suggest you add more images to complete the manuscript. It may be of interest to add orthogonal radiographic images of the chest, although no abnormalities were observed.

We have added orthogonal radiographic images of the chest following the Reviewer’s suggestion.

Please add the model of the ultrasound machine and the probes (frequency) you use in the text.

We have added the requested information, as follows: “…an echocardiographic examination was performed using an ultrasound unit equipped with multifrequency 1-4 MHz phased-array transducer (MyLabTM Eight, Esaote, Genova, Italy)…”.

Also add Transesophageal echocardiography images

We have added two transesophageal echocardiographic images.

Reviewer 3 Report

The case report reported by Bufalari et al is well written, well performed and well explained work. The reviewer does not have any request. I think that the stream of procedures were all legit, well described and ,especially, well performed. 

-were there any cytological examination of the thoracic effusion maybe? in case that would also definitely add some more information for the readers

congratulations for this nice case report

Author Response

We thank the Reviewer for her/his encouraging comments. We have addressed the comment raised and we hope that our response below will satisfy the Reviewer.

The case report reported by Bufalari et al is well written, well performed and well explained work. The reviewer does not have any request. I think that the stream of procedures were all legit, well described and ,especially, well performed.

-were there any cytological examination of the thoracic effusion maybe? in case that would also definitely add some more information for the readers

Pleural and pericardial effusion was not present in our case. Therefore, no cytological examination has been performed.

congratulations for this nice case report

We thank again the Reviewer for her/his comments.